# Systemically Administered, Target-Specific, Multi-Functional Therapeutic Recombinant Proteins in Regenerative Medicine

**DOI:** 10.3390/nano10020226

**Published:** 2020-01-28

**Authors:** Tero A.H. Järvinen, Toini Pemmari

**Affiliations:** Faculty of Medicine & Health Technology, Tampere University, FI-33014 Tampere, Finland & Tampere University Hospital, 33520 Tampere, Finland

**Keywords:** vascular homing peptide, cell penetrating peptide, angiogenesis, vascular heterogeneity, fibrosis, targeted delivery, decorin, transforming growth factor-β (TGF-β), bystander effect, CendR peptide, tissue regeneration, regenerative medicine, hypoxia, neuropilin-1, stem cell

## Abstract

Growth factors, chemokines and cytokines guide tissue regeneration after injuries. However, their applications as recombinant proteins are almost non-existent due to the difficulty of maintaining their bioactivity in the protease-rich milieu of injured tissues in humans. Safety concerns have ruled out their systemic administration. The vascular system provides a natural platform for circumvent the limitations of the local delivery of protein-based therapeutics. Tissue selectivity in drug accumulation can be obtained as organ-specific molecular signatures exist in the blood vessels in each tissue, essentially forming a postal code system (“vascular zip codes”) within the vasculature. These target-specific “vascular zip codes” can be exploited in regenerative medicine as the angiogenic blood vessels in the regenerating tissues have a unique molecular signature. The identification of vascular homing peptides capable of finding these unique “vascular zip codes” after their systemic administration provides an appealing opportunity for the target-specific delivery of therapeutics to tissue injuries. Therapeutic proteins can be “packaged” together with homing peptides by expressing them as multi-functional recombinant proteins. These multi-functional recombinant proteins provide an example how molecular engineering gives to a compound an ability to home to regenerating tissue and enhance its therapeutic potential. Regenerative medicine has been dominated by the locally applied therapeutic approaches despite these therapies are not moving to clinical medicine with success. There might be a time to change the paradigm towards systemically administered, target organ-specific therapeutic molecules in future drug discovery and development for regenerative medicine.

## 1. Drug Delivery in Regenerative Medicine

Intensive research during the past few decades has led to the identification of the key molecules, growth factors, cytokines and chemokines, for tissue regeneration after injuries [1,2]. As most of the top-selling drugs are recombinant proteins in the modern world, it was anticipated that those powerful molecules could be expressed as recombinant proteins and used as drugs in humans to augment the repair of tissue injuries. Unfortunately, their clinical use to enhance tissue regeneration in humans has been scarce [1,3,4], as the molecules have failed in clinical trials. There are several biological reasons for the failure: the instant degradation of locally applied proteins in the protease-rich environment present in the injured tissue, the inability to retain small (growth factors, chemokines and cytokines are very small in size) recombinant proteins at the site of injury, poor tissue penetration, and side effects [4,5,6]. Severe systemic toxicity, increased cancer risk and biological effects outside of the target cells were the side effects that halted the human clinical trials testing cytokines and growth factors. These biological issues illustrate the major roadblocks that need to be addressed for biological protein-based therapeutics to be successful as a therapy in humans. Taken together, future pharmaceutical treatment options with recombinant proteins should focus on (i) reducing or eliminating the degradation of the recombinant protein-drugs in the inflammatory milieu of the injured tissue and/or (ii) to extend their retention time at the wound site [7]. Besides to these issues, most injuries are inaccessible for local delivery or the exposure of injured site is not desired for biological reason (such as fractures) or involve multiple tissues injured simultaneously. These situations naturally call for a systemic administration. 

In this regard it is actually quite stunning that almost all efforts with recombinant proteins aimed to enhance tissue regeneration are based on their local administration [4,5,6,8]. Although systemic drug administration of both recombinant proteins and conventional drugs is the only drug delivery mode used for the vast majority of human diseases, systemic administration has not been considered as a viable option in the treatment of tissue injuries due to the lack of efficiency and safety. These concerns are justified as only a small fraction of systemically administered drug reaches its desired location in the body and the side effects related to the therapy as well as severe consequences such as the increased cancer risk could be encountered elsewhere in the body [7]. In addition to these safety concerns, large drugs such as therapeutic antibodies have poor tissue penetration and therefore might not reach the intended target [9,10,11,12]. Target organ-specific drug delivery obtained by the combination of vascular homing peptides and functional protein domains, such as cell penetrating peptides proficient in penetrating cells and tissues, could solve these problems. They could provide the means for the selective accumulation of the systemically administered therapeutics in the injured tissue [12,13,14,15,16,17]. 

### Vascular Heterogeneity-Zip Code-System in Vasculature Enables Tissue-Specific Drug Delivery

One of the goals of the modern pharmaceutical treatment is to be as target specific as possible; drugs should be highly active against the disease, while having as few side effects in the healthy parts of the body as possible [18]. This goal is usually obtained by developing drugs that act on molecules selectively over-expressed by the cells in the diseased organ. The broadening understanding of the biology could provide new means to convert conventional drug to a target site-specific by a targeted delivery to the desired location. The vascular system provides a natural platform for doing that. The expanding understanding of the molecular composition of blood vessels has shown that each tissue leaves its mark (on molecular level) on the lumen of the blood vessels in the given tissue. The blood vessels in each tissue have molecular structures (“molecular fingerprint”) on their lumen that are unique for the given tissue, essentially forming a system that is very similar to postal codes in society (“vascular zip codes”) [19,20] (Figure 1). Moreover, various diseases induce the expression of disease-specific, unique molecular signatures on the vasculature, essentially providing an appealing target for the disease specific delivery of systemically administered drugs [19,20]. This is particularly evident for diseases like cancer and tissue injuries; both of which are associated with tissue hypoxia, which in turn induces new blood vessel growth by angiogenesis [20]. The angiogenic blood vessels are structurally and molecularly different from the dormant blood vessels elsewhere in the body and provide an accessible and abundant target for the organ-specific delivery of therapeutics [19,20] (Figure 2). These organ- and disease-specific vascular zip codes can be targeted by systemically administered affinity ligands that seek out, i.e. “home”, to their target by binding to the corresponding vascular zip code, i.e. receptor on the lumen of the blood vessel [19,20,21] (Figure 2).

A robust technical approach utilized for discovering ligands (vascular homing peptides or antibodies) for vascular zip codes has been in vivo phage display, which has been used successfully to discover numerous efficient homing peptides and antibodies. Pasqualini and Ruoslahti introduced the in vivo biopanning i.e. a phage display method where the phage library is injected into circulation, i.e. spreading the library on the luminal side of the vasculature of an individual instead of spreading it on a cell culture plate [21]. Their work revealed that some peptides concentrate to given tissue, which means that vasculature has tissue-specific features and these features are detectable by small peptides [21]. Since that, tens of tissue homing peptides have been discovered. Some of them now have their receptor identified, while the homing mechanism of the others is still unraveled [19]. It has emerged that some of the peptides are capable of homing only but some of them can penetrate to their target tissue [19]. The homing devices discovered by in vivo phage display highlight the remarkable diversity that exists in the blood vessels between different tissues and the pharmaceutical opportunities that exist in our body. 

### Ligands Targeting Injured Tissues

Tissue injuries, fractures, skin wounds, tendon and skeletal muscle ruptures are a very common major medical problem, as we have no pharmaceutical means to accelerate or enhance the natural healing process nor the potential surgical procedure. Tissue injuries also provide an excellent target for the ligand-mediated delivery of systemically administered therapeutics [18]. The first actual regenerative sign in the repair process after an injury is the massive angiogenesis induced in the injured area by the hypoxia. The injured area is hypoxic due to the disrupted circulation in the damaged tissue [18,22,23]. The surviving cells and the inflammatory cells that have extravasated to the injured area experience hypoxia in oxygen deprived conditions. When the cells experience hypoxia, it stabilizes the transcription factor hypoxia inducible factor-1α (HIF-1α), which accumulates in the cell nucleus to direct gene expression towards helping one to address hypoxia [24,25]. Active HIF-1α signaling triggers the production of large number of angiogenic growth factors, among them vascular endothelial growth factor-A (VEGF-A) [24,25]. These soluble growth factors direct the sprouting of the new blood vessels, i.e. angiogenesis, to deliver oxygen and address the hypoxia in the tissue [24]. The abundance of angiogenic blood vessels in the granulation tissue (the loose connective tissue produced in the early phase of the healing period) is so abundant that the word “granulation tissue” is actually derived from the granular appearance of the angiogenic capillaries. Thus, the injured area is essentially made out of the tiny angiogenic capillaries in the proliferation phase of the healing. This thick microvascular bed essentially fills up the injured area and presents large number of molecular targets, i.e. sprouting angiogenic blood vessels expressing an overwhelming number of angiogenesis-related “molecular fingerprints” for systemically administered homing ligands to bind and deliver therapeutics to the regenerating tissue in target-specific fashion (Figure 2). To obtain an “universal vehicle” for targeting tissues undergoing repair and regeneration in any given tissue, we screened a random peptide library (1.0 × 10^9^) by in vivo phage display for injury-homing peptides against excision wounds in skin and lacerations or ruptures in tendon at the peak of angiogenesis [26]. The screening of the random peptide library (CX7C-library) yielded peptides that homed to the angiogenic blood vessels in the newly formed granulation tissue [26]. The most efficient homing peptide among the peptides has the amino acid sequence CARSKNKDC (cysteines at the end of peptide make disulfide bonds and create a cyclic peptide). The peptide was dubbed “CAR” based on the beginning of its amino acid sequence. The analysis of the CAR sequence reveals a classical heparin-binding motif that shows high homology with the heparin binding domain of bone morphogenetic protein-4 (BMP-4), a well-established angiogenic growth factor. CAR peptide accumulates in the skin and tendon wounds and injuries at levels that are as much as 100–200 times higher than the levels of control peptides, and the homing is injury-specific; no CAR is detected in normal tissues. Importantly, CAR is a potent cell and tissue penetrating peptide since it effectively accumulates deep in the tissue parenchyma and internalizes to cultured cells [26]. CAR binds to the cell surface heparan sulfate proteoglycans (HSPGs) and uses the HSPGs as receptors for cell binding and penetration into cells and target tissues [26]. The subsequent research has demonstrated that in addition to angiogenesis in injured tissues undergoing repair and in human tumor xenografts [26,27], CAR also recognizes reactive vasculature in different experimental disease models of inflammatory diseases. It shows remarkable homing specificity (no homing to respective normal tissue) to pulmonary arterial hypertension (PAH), myocardial infarction, abdominal aorta aneurysm (AAA) and dystrophic muscle lesion in muscular dystrophies [27,28,29,30,31]. 

Although the target site-specific delivery of the therapeutics is an emerging field in regenerative medicine, quite many different practical applications for vascular homing peptides are already emerging and the field is exploding with new discoveries. Some of these regenerative medicine-intended homing peptides are more general, i.e. can be applied to all injuries, such as identifying peptides capable of binding to the angiogenic or reactive (or towards diseases with inflammatory component) blood vessels forming at the injured tissues [26], or peptides binding to the extracellular matrix (ECM) collagen exposed in inflammatory conditions [32]. Regenerative medicine intended vascular homing peptides have been described towards specific clinical situations, such as targeting blood clots that form after an acute, traumatic tissue injury [33] or towards angiogenetic blood vessels at early [34] and late stage of wound healing when granulation tissue matures to scar tissue [26]. In addition to these, homing peptide targeting tissue factor after blood vessel rupture has been used to deliver amphiphile nanofibers to control hemorrhage [35], targeted axonal import peptide delivered therapeutic recombinant proteins to transected nerves after spinal cord injury [36]. Homing peptides that seek out burn injury in intestine [37,38] or deliver diagnostic or therapeutic nanoparticle to brain injury after a traumatic breakage of the blood-brain barrier [39,40] have been described. There are also approaches for targeting systemically administered molecules to bone tissue or fracture by homing devices (homing peptides and aptamer) that are not blood vessel specific but bind either to tartrate-resistant acid phosphatase deposited by osteoclasts or to hydroxyapatite deposited by osteoblasts [41,42,43]. Vascular homing peptides targeting bacterial infections [44] or simply illuminating neural structures during surgery have also clear regenerative medicine intentions [45,46] (Table 1).

### Systemically Administered Anti-Fibrotic Molecule, Car-Decorin

To utilize the homing properties of CAR in a therapeutic way, we generated a multi-functional therapeutic recombinant protein in hopes of creating a targeted anti-fibrotic molecule [47]. In this multi-functional recombinant protein CAR peptide is expressed together with the decorin (DCN) core protein (CAR-DCN) [47,48] (Figure 3). CAR functions (1) as a homing device in the fusion protein that delivers the therapeutic molecule to the target, (2) mediates cell and tissue penetration, and (3) attaches its payload to the target cells through HSPGs [47] (Figure 3). DCN, in turn, functions as a therapeutic molecule in the multi-functional fusion molecule. DCN is a member of the small leucine-rich proteoglycan (SLRP) family of ECM proteins. Owing to its physical interactions with the collagen fibers in the ECM, i.e. DCN decorates collagen fibers in the ECM, the proteoglycan was named decorin [49]. In addition, being a structural component of the ECM, DCN influences cellular functions such as proliferation, spreading, migration, differentiation and regulates inflammation [50,51,52]. DCN has received much attention as a therapeutic agent because of its anti-fibrotic, -inflammatory and -cancer effects [53,54,55]. The anti-fibrotic function of DCN is related to its property of being a natural inhibitor of TGF-β, a growth factor responsible for scarring and fibrosis [50,52,56,57]. The scar-inducing activities of TGF-β1 are mediated by the connective tissue growth factor (CTGF/CCN2) and the epidermal growth factor (EGF) family receptors (ERBBs) [58]. DCN also neutralizes CCN2 [59], ERBBs [60,61,62] and myostatin [63,64], an important contributor to scarring in several organs. Thus, a single DCN molecule can neutralize multiple fibrosis-inducing growth factors simultaneously as the binding sites for these growth factors exist in different parts of DCN molecule [54]. In other words, it inhibits scar and fibrosis formation by multi-tasking. 

Systemically administered CAR-DCN homes to skin wounds, abdominal aorta aneurysms and dystrophic muscle lesions substantially better than the native DCN. This can be considered as a remarkable feat in the light of studies demonstrating that the DCN core protein homes to angiogenesis [65,66] and has been used as a delivery vehicle for other therapeutics [67]. Most recently, a collagen binding peptide derived from the DCN sequence was used as a delivery vehicle for anti-inflammatory and –fibrotic therapeutics to inflammatory diseases [32]. Furthermore, on top of these homing features DCN possesses, the naturally occurring glycosaminoglycan (GAG) in DCN can bind to α2β1-integrins on angiogenic endothelial cells [68]. Taken together, the substantially better homing of CAR-DCN than the native DCN illustrates the advantages offered by a vascular homing peptide even for a therapeutic protein that has an inherent ability to home to target organ.

Systemically administered CAR-DCN is substantially better than native DCN in scar prevention and tissue regeneration [47], attenuating the progression of the aorta aneurysm progression [30], and reducing the severity of the muscular dystrophy (Figure 3). The molecular mechanism behind the enhanced biological activity of CAR-DCN is not only related to homing, i.e. the better accumulation of the therapeutic molecule at the site of the injury or the disease. We have demonstrated that CAR-DCN is substantially more active than the native DCN in neutralizing TGF-β, i.e. suppressing its bioactivity [47]. It was demonstrated that DCN binds to all isoforms of TGF-β [57], CAR-DCN is significantly more active than DCN in neutralization of TGF-β1 and -β2 but shows no activity against TGF-β3 [47]. The differential inhibitory activity of CAR–DCN against the TGF-β isoforms could be clinically highly relevant; TGF- β1 is the growth factor responsible for scar formation, while TGF-β2 enhances the profibrotic activity of TGF-β1 [69]. TGF-β3, in turn, could have substantial anti-scarring activity. A molecular basis for this selectivity could be the different HSPG-binding properties of CAR-peptide and the TGF-β isoforms. It has become apparent that some isoforms of TGF-β bind HSPG [70,71,72]. CAR-peptide, TGF-β1 and -β2 bind HSPG, but TGF-β3 does not. This effect may enhance the biological activity of CAR-DCN and provide unique selectivity to the molecule by bringing CAR-DCN to the proximity of scar-inducing isoforms of TGF-β, namely -β1 and -β2, that seek HSPG binding [71] (Figure 3). Interestingly, a large number of naturally occurring TGF-β/bone morphogenetic protein (BMP) antagonists are also heparin binding proteins [71]. Thus, CAR peptide may have converted DCN into a selective inhibitor of scar-inducing isoforms of TGF-β through the selectivity obtained by HSPG binding [47,54]. The re-engineering of the therapeutic proteins by providing them with an additional heparin binding domain could be an emerging theme in tissue engineering and regenerative medicine as a similar approach to the one used in the engineering of CAR-DCN, has been utilized more recently in generating super growth factors [8,73]. The super growth factors were created by fusing a growth factor with a heparin binding domain from placental growth factor (PLGF). Whereas the CAR peptide has homology with the heparin binding domain of BMP-4 [26], the addition of the heparin binding domain of PLGF makes the chimeric super growth factors more active than the native ones by affording them an enhanced heparan sulfate-dependent cell binding and presentation to the growth factor receptor [8]. Furthermore, it was recently demonstrated that the addition of cell surface HSPG binding domain, i.e. syndecan-binding domain, to growth factors induces tonic growth factor signaling [73]. In line with the syndecan-binding domain enhancing growth factor signaling, it has been reported that the use of locally applied, fibrin-conjugated peptides with heparin binding domains enhances wound healing [74,75]. The authors attributed the beneficial effects of the biomaterial on would healing on growth factor retention in the wounds by binding to the heparin-binding domain [74,75]. On the other hand, it was recently demonstrated that some of the heparin-binding growth factors bound too strongly to the HSPGs in the ECM. Their biological activity is actually restricted by the strong heparin and ECM binding. Tissue regeneration was enhanced by the application of an enzyme, sulfatase, that cleaves growth factor docking sites from HSPGs and released the trapped growth factors to increase their bioavailability [72]. In line with that study, a chimeric fibroblast growth factor 21 (FGF21) was generated by substituting the thermally labile and low receptor affinity core of an FGF hormone (FGF21) with an HSPG binding-deficient endocrinized core derived from a stable high receptor affinity paracrine FGF [76]. The chimeric FGF21 is metabolically more potent and longer acting than the native FGF21 [76,77]. Altogether, these findings point that the biological activity of many proteins can be modified by making them chimeric with the introduction of right HSPG binding domain.

There are large number of systemically administered, target-specific multi-functional recombinant fusion proteins that contain both targeting and therapeutic domains developed against cancer or autoimmune diseases. Some of these could also be applied to regenerative medicine with a very specific indication. Large number of such molecules carry either anti- or pro-inflammatory protein in hopes of deciphering immune response [32,78,79,80,81,82,83,84]. Recently, a homing peptide targeting tumor ECM delivered tumor necrosis factor-α (TNFα) to ECM rich regions in tumors and caused immune cell infiltration and subsequent immune cell-induced ECM breakage [79]. Theoretically, similar approaches inducing “immunological reaction towards connective tissue” could be applied to situations where scarring or fibrosis has ensued, and the breakage of the fibrosis is desirable without side effects in the rest of the body. Most recently, the technology to target powerful cytokines was further refined by generating novel “designer cytokines” that have reduced systemic toxicity to one afforded by the selective delivery to target organ [85,86]. Namely, these “designer cytokines”, TNFα and interferon-γ (IFNγ) are delivered to the angiogenic vasculature by a separate vascular targeting domain [85]. In addition to that, they have a mutated receptor binding site in the therapeutic molecule, i.e. TNFα and IFNγ, to reduce their biological activity [85,86]. The reduced biological activity, in turn, reduces the systemic toxicity of the molecules and makes them viable, tolerated and safe drug option for systemic administration. Selective vascular stabilization of the angiogenic blood vessels (that possess poor perfusion) was obtained by targeted delivery of therapeutic cytokine, TNF family member LIGHT (also known as TNF Family member 14) [87,88]. The vascular stabilization of angiogenic neovessels could be also a viable option for regenerative medicine to improve blood perfusion and nutrient delivery, not only for acute injuries with high metabolic demand, but also chronic injuries where hypoxia and angiogenic, nonfunctional blood vessels persist [89]. Furthermore, biphasic homing peptides that target two different ECM molecules, tenascin-C and fibronectin, in tumor ECM [90,91], could also be used for regenerative medicine, as the granulation tissue in acute tissue injuries is rich in these two ECM molecules [92]. 

Combining a short vascular homing peptide (and potential cell penetrating motifs) as part of a large recombinant fusion protein makes it possible to circumvent two major pharmacological disadvantages of peptides, the short half-life in circulation and the low binding affinity towards the target receptor [18]. Thus, it is not striking that multiple multi-functional therapeutic recombinant fusion proteins derived from inserting multiple functional “domains” together, i.e. vascular targeting element and a therapeutic molecule, have been described in the biomedical literature and the most advanced ones have already moved to clinical trials (Phase III) in humans. Most of these molecules are in the field of cancer research. A concern when a vascular homing peptide is used as a part of a therapeutic protein made out of the multiple domains is the potential immunogenicity of the resulting fusion protein. The vascular homing peptides are generally short (<10 amino acids), and as such, are unlikely to be immunogenic. For example, they are substantially shorter that the complementary determining regions (CDRs) inserted into therapeutic antibodies, which usually do not cause immune reaction and have a reputation as safe drugs. One also needs to take into account that vascular homing peptides sometimes share sequence homology with parts of human and mammalian proteins, as is the case with CAR peptide, which shows high degree homology with the heparin-binding domain of BMP-4. This feature most probably reduces immunogenicity, as the native domain already exists in the host body [18].

### Bystander Effect–A Novel Approach to Drug Delivery 

The most remarkable breakthrough in the field of targeted therapeutics is the demonstration that the systemically administered drugs can also be converted to target organ or disease specific by co-administering the drug simultaneously with certain homing peptides without a need to bind or chemically couple the peptide to the drug [19,93] (Figure 4). The mechanism is the activation of a trans-tissue transport pathway by the vascular homing peptide [19,94,95]. The drug is essentially “sucked” into this pathway and transported into the target tissue of the homing peptide (bystander effect) (Figure 4). The bystander effect was initially discovered with an integrin-binding vascular homing peptide that has tumor-penetrating properties [93,96]. In addition to the actual homing peptide sequence, the tumor-specific cell and tissue penetrating homing peptides contain a consensus (cryptic) motif R/KXXR/K, with an arginine (or rarely lysine) residue at the C-terminus, thus called the C-end Rule (CendR) sequence (Figure 5). The CendR-peptide binds to its integrin receptor, which is specific for tumor endothelium, and becomes cleaved by protease to expose a cryptic CendR motif within peptide. Once CendR motif is exposed (in the C-terminus) it can bind to another receptor, neuropilin-1 (NRP-1) [93,96]. The NRP-1 binding by CendR-motif activates the transport pathway (CendR pathway), which transports the peptide and a co-administered (or coupled) drug from vessels deep into the target tissue [19,95] (Figure 4 and Figure 5). Even large molecules such as antibodies and nanoparticles can be transported in a tumor-specific fashion using the tumor-penetrating peptides without coupling them to the peptide [93,97].

So far, several tumor homing peptides with CendR domain that are capable of inducing the bystander effect have been reported. In addition to the CendR peptides, CAR peptide is the only known peptide without the CendR domain that is capable of inducing a bystander effect on drugs administered simultaneously, but not conjugated to the peptide [18,28] (Figure 6). CAR peptide has homing and cell and tissue penetrating properties similar to those in CendR-peptides. The main difference is that the internalization of heparin-binding CAR peptide is mediated by HSPGs and not by NRP-1 [98]. There is evidence that after initial peptide binding to either NRP-1 or HSPG, the pathways merge and utilize the same “channel” through the cells for drug delivery to the tissue [98].

The CAR peptide induced bystander effect has been demonstrated in pulmonary arterial hypertension (PAH) as a way to get tissue selective vasodilation [28] (Figure 6). A fundamental problem with the pharmaceutical treatment of PAH is that the disease is very resistant to vasodilation induced by blood pressure lowering drugs. As no anti-hypertensive drug selective for the pulmonary circulation exists, the anti-hypertensive drugs need to be used in high doses in PAH. This causes side-effects related to high drug doses such as an unwanted drop in systemic blood pressure (hypotensive shock) and limits the utility of these drugs in the treatment of the lethal disease. Co-administering CAR peptide with blood pressure lowering drugs converts these drugs to tissue specific in their action, i.e. they only lower blood pressure in lung vasculature [28]. The mechanism is the following: CAR peptide homes to the inflammatory pulmonary blood vessels in PAH (but not to normal lung vessels) and penetrates deep into the vessel wall in experimental models of PAH [28]. It induces the bystander effect on drugs administered simultaneously in PAH, i.e. it increases the accumulation of the co-administered drugs in the vessels affected by PAH. The increased accumulation of drugs induces pulmonary vasodilation without affecting blood pressure on the systemic side of the circulation [27,28] (Figure 6). Drug doses as low as 10% of the dose used when drugs are administered alone (conventional drug administration) were effective in lowering pulmonary blood pressure in PAH, while having little or no effect on the blood pressure on the systemic circulation [28] (Figure 6). Taken together, CAR peptide induced targeted delivery of blood pressure lowering drugs to PAH. The selective vasodilation in pulmonary circulation could be the next generation drug treatment in PAH; one that is effective but does not have the side effects associated to conventional pharmaceutical approaches.

The activation of the endocytic transcytosis and trans-tissue transport pathway by the CendR and CAR peptides that triggers the bystander effect for drug delivery reveals the importance of developing targeting devices that not only deliver the therapeutic molecules to the target organ but also provide a mechanism for cell and tissue penetration. Proper cell penetration guarantees that the drug indeed accumulates at the site of the disease and does not only stick to the endothelium in blood vessels. Cell penetration afforded by CendR-pathway also solves targeting restrictions related to low number of receptors in the blood vessels of the target organ [10] as the targeting peptide is constantly shuttled to the parenchyma rendering the receptor available to binding again. Another similar innovative ways to deliver molecules deep in tissue parenchyma are antibodies that bind to caveolae and are pumped into the target organ [99,100] and having both cell binding and internalization domains in one molecule [17].

### Targeted Nanoparticles And Stem Cells for Regenerative Medicine 

A sustained drug release can be obtained by packaging large amounts of therapeutic molecules into nanoparticles. Any particle can be called as a nanoparticle if its diameter is 1000 nm or less. Various nanoparticles have been introduced as potential drug carries: organic, liposomes, micelles, nanoerythrosomes, dendrimers, carbon-based, fullerenes, or inorganic nanoparticles, silicon nanoparticles, metal or quantum dots [16,19]. These particles can be loaded with drugs to offer advantages like prolongation of drug release, tissue deposition for efficient drug absorption and simultaneous delivery of multiple therapeutic drugs [101,102,103,104,105,106]. Nanoparticles offer an option to modify them on their surface, by applying targeting elements to the surface of the particles. The modified particles can be targeted to the diseased vasculature [13,16,19,102,104,105,107,108,109]. This could be especially important because many nanoparticles (especially the ones made from inorganic materials) are rapidly removed from the circulation by the reticuloendothelial system [19]. Thus, the delivery of the nanoparticle to the target organ by a vascular homing peptide could substantially increase the amount of the nanoparticle-loaded drug accumulation in the desired target [19]. 

Similar, desirable, selective vasodilation to one that CAR peptide induces through bystander effect in PAH, has also been achieved with various types of nanoparticles (erythrocytes, liposomes, micelles) coated with CAR peptide in different experimental models of PAH. These CAR peptide targeted nanoparticles are loaded with an anti-hypertensive drug that is released slowly from the nanoparticles [102,103,105,110]. As the nanoparticles are targeted by CAR peptide to pulmonary circulation in PAH, the drug release takes place in the target organ and the vasodilation is organ specific. The nanoparticle approach eliminates a major limitation of the CAR peptide co-administration [28], short half-life of the peptide and the drug and provides extended selective vasodilation in PAH. CAR peptide delivered nanoparticles that contained anti-hypertensive drugs induced selective vasodilation in pulmonary vasculature (no concomitant drop of blood pressure in the systemic circulation) and extended the duration of selective vasodilation up to 15 times from the one obtained by conventional drug administration [102,104,105,107,108,110,111]. Other, recent examples of targeted nanoparticles are the demonstration of targeting nanoparticles loaded with antibiotics to infection foci for an enhanced anti-bacterial activity [44] and macrophage-specific nanoparticles that introduce an oligonucleotide directly to the cytosol for gene knockdown to enhance the macrophages´ capacity to combat staphylococcus aureus infection [112]. Placenta targeting nanoparticles carrying insulin-like growth factor 2 (IGF2) improved fetal weight in a model of fetal growth restriction [113] and a selective delivery of nanoparticle carrying nitric oxide releasing drug to uteroplacental vasculature rescued impaired placental perfusion [114].

In addition to applying vascular homing peptide technology to delivery of nanoparticles, the stem cells have been targeted with them as well [29,115]. Different homing peptides were used to deliver mesenchymal stem cells, which were “painted” with multiple copies of the peptides on their cell surface, to obtain improved accumulations of therapeutic stem cells in the target organ [29,31]. This can be obtained by synthesizing the homing peptide with a palmitic acid tail to facilitate cell membrane integration [29,116]. In addition to stem cells, exosomes secreted by stem cells have been delivered to their desired target by homing peptides [117]. 

## 2. Future Perspectives

Ligand-mediated targeting of systemically administered pharmaceutical agents shows a promise in the treatment of different diseases. The approach is also emerging in the field of regenerative medicine for tissue injuries, illustrated by an explosion of the scientific discoveries and papers describing different applications of the technology. The homing peptide CAR has demonstrated that a target-specific delivery of therapeutic molecules can be achieved as a part of multi-functional recombinant protein, painting nanoparticles or stem cells with the homing peptide, as well as without any physical attachment of the drug to the peptide through the so-called bystander effect. The outcome of the targeted delivery is the improved specificity and efficacy of the targeted drug. Numerous studies have demonstrated that the homing peptide can enhance the biological activity of the therapeutic recombinant protein itself and innovative strategies to re-engineer the recombinant multi-functional proteins further have been described. Although the field of regenerative medicine has been dominated by the locally applied therapeutic approaches, these therapies have not moved to clinical medicine. There might be a time to change the paradigm towards systemically administered, target organ-specific therapeutic molecules in future drug discovery and development for regenerative medicine. 

## Figures and Tables

**Figure 1 nanomaterials-10-00226-f001:**
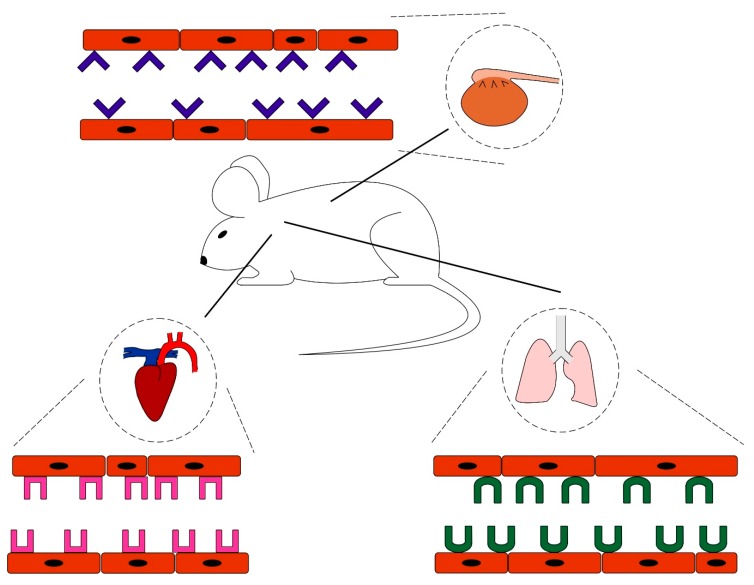
The vascular zip codes. Each tissue has its own unique molecular structure in the luminal side of its vessels. These molecules function as receptors for tissue homing molecules such as peptides. Combining a homing peptide with another molecule creates a compound that can home to its target tissue. Homing diminishes the amount of the drug needed and helps to avoid side effects of the medication.

**Figure 2 nanomaterials-10-00226-f002:**
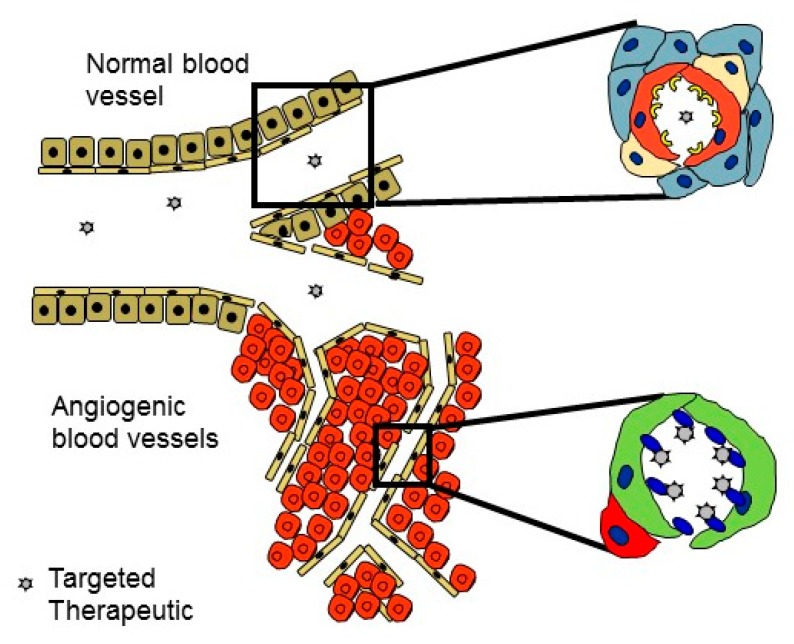
Target organ-specific delivery of therapeutics in regenerative medicine. The angiogenic blood vessels that form in regenerating tissues after injury are structurally and molecularly different from the dormant blood vessels elsewhere in the body. They provide an accessible and abundant target for the organ-specific delivery of therapeutics during the repair of the tissue injuries. They can be targeted by vascular homing peptides that bind to their receptor selectively expressed only in angiogenic blood vessels. Drugs administered systemically can be converted to target tissue-specific by coupling them to the vascular homing peptide.

**Figure 3 nanomaterials-10-00226-f003:**
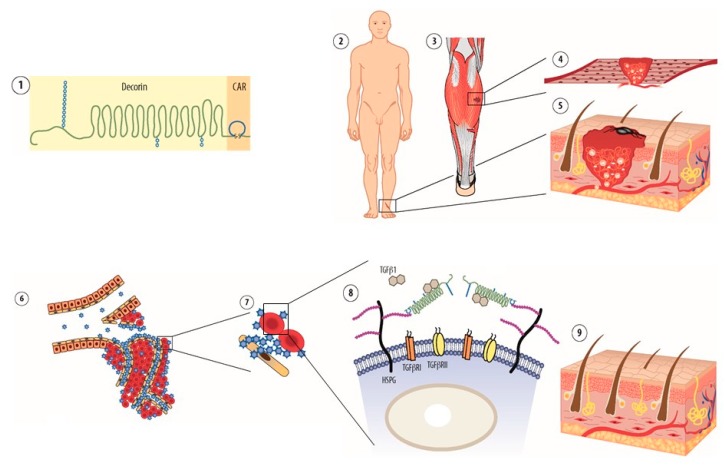
Mechanism of action of multi-functional re-engineered decorin in inhibition of scar formation. Schematic presentation on recombinant fusion protein consisting of decorin (DCN) and the vascular homing and cell penetrating peptide CAR. CAR-decorin ① is a multifunctional biotherapeutic that inhibits numerous growth factor signaling pathways involved in fibrosis. The systemically administered molecule is targeted by CAR peptide to the inflammatory or angiogenic vasculature in any organ of the body ②. The CAR homing peptide binds to its receptor (“zip code”) in angiogenic or inflammatory vasculature ② and as cell penetrating peptide, it delivers the fusion molecule deep in the target organ parenchyma ③. The CAR peptide then binds to heparin sulfate proteoglycans on the cell surface of the stromal cells ③. This binding facilitates the neutralization of scar forming isoforms TGF-β1 and TGF-β2 by the therapeutic part of the molecule, DCN ④. This mechanism results in a therapeutic response, i.e. reduction of scar formation in skin wound. Picture by Helena Schmidt (With permission from Suomalainen Lääkäriseura Duodecim 2011).

**Figure 4 nanomaterials-10-00226-f004:**
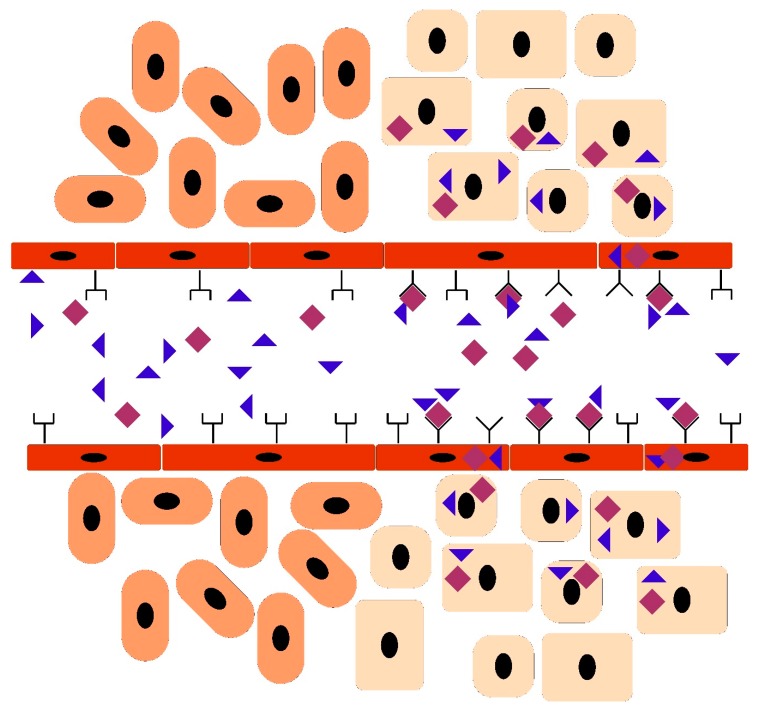
The bystander effect enables delivering a therapeutic into the tissue without a covalent link between the therapeutic and the homing peptide. The drawing represents the homing of a peptide (diamond) to its target tissue (cuboidal cells) and the bystander effect created by the peptide. The tissue on the right has the receptor for the homing peptide (Y-shape). When the peptide binds to its receptor, it opens a pathway for the therapeutic molecule (triangle) to pass through the vascular wall endothelium and enter the tissue parenchyma.

**Figure 5 nanomaterials-10-00226-f005:**
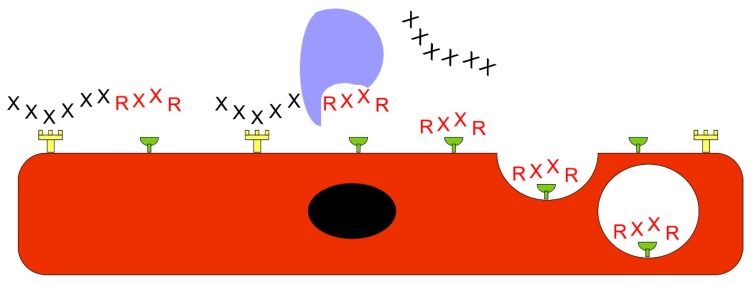
The function of the C-end Rule sequence. Peptides containing the C-end Rule (CendR) sequence are capable of penetrating to tissues and delivering cargo. Typically, the CendR sequence is cryptic and needs to be exposed. The receptor specific for the target tissue (fork) first binds the homing peptide enabling its cleavage by an enzyme (blue), which results in the exposure of the CendR sequence (RXXR). The CendR sequence then binds to another receptor (half a circle), neuropilin-1 (NRP-1), which triggers the cell membrane to form a vesicle containing the peptide and other molecules close to it. The vesicle is then transferred through the cell to let its contents enter to the tissue parenchyma.

**Figure 6 nanomaterials-10-00226-f006:**
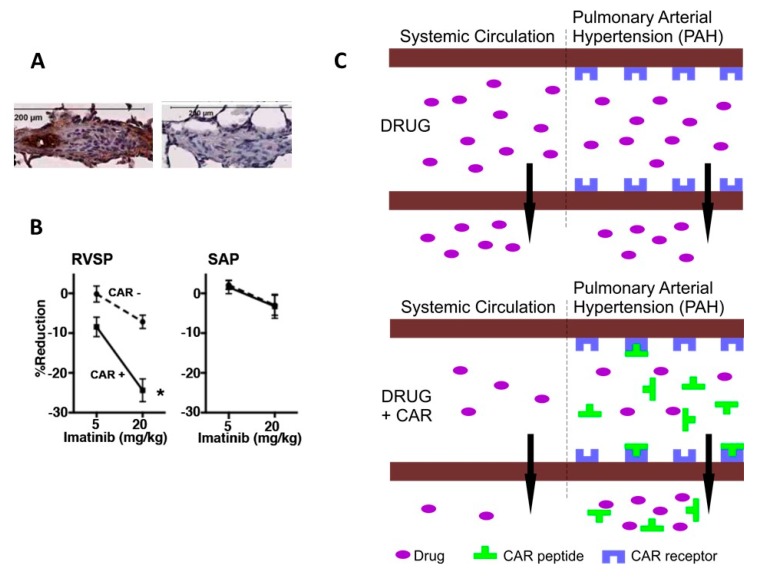
Vascular homing peptide induced tissue-selective vasodilation in pulmonary arterial hypertension (PAH). (**A**) Vascular homing and cell penetrating peptide CAR homes to pulmonary arteries in disease-specific fashion in experimental models of PAH [27,28]. A little bit of control peptide (mutant CAR peptide) binding can be seen to similar blood vessel in PAH. (**B**) Effect of CAR (0.3 mg/kg) and Rho-kinase inhibitor Y27632 (1 mg/kg) mixture on right ventricle (RVSP, right ventricle systolic pressure) and left ventricle (SAP, systemic arterial pressure) systolic pressure in PAH. The CAR/Y27632 combination treatment induced a marked pulmonary-specific vasodilation RVSP with only a minimum effect on SAP in PAH. (**C**) Schematic presentation of the “bystander effect”, i.e. target organ-specific drug delivery, in PAH. Reproduced from [28] with permission from Elsevier 2017.

**Table 1 nanomaterials-10-00226-t001:** Vascular homing peptides described for regenerative purposes in tissue injuries.

Peptide	Sequence	Target	Receptor	Therapeutic application	Reference
CAR	CARSKNKDC	angiogenesis (early state) and reactive vasculature in vast number of inflammatory diseases	heparan sulfate proteoglycans	recombinant proteins	[26,30,47]
				stem cells	[29]
				unconjugated drug delivery (bystander effect)	[28]
				nanoparticles	[26,27,28,96,97,98,99,100,102,103,105,106]
CARG	CARGGLKSC	*Staphylococcus aureus* infected tissue		nanoparticles	[44]
c-RGDfK	CRGDFC	Angiogenesis	α_v_β3 integrin	nanoparticles	[34]
CAQK	CAQK	acute brain injury	chondroitin sulfate	siRNA	[39,40]
CH6	see ref. [40]	osteoblasts		lipid nanoparticle	[42]
CLT1	CGLIIQKNEC	blood clots	chloride intracellular channel 3 and integrin α5 (β1)		[33]
CRK	CRKDKC	angiogenesis (late stage)		stem cells	[29]
				nanoparticles	[26,104]
Np41	NTQTLAKAPEHT	nerves	laminins α2, α4	identification of nerves during surgery	[45,46]
poly-Asp	DDDDDDDC	bone	hydroxyapatite	nanoparticles	[43]
RTL	RTLAFVRFK	ruptured blood vessel	tissue factor (TF)	nanofibers	[35]
TAxI	SACQSQSQMRCGGG	transected nerves		recombinant proteins	[36]
TBP	TPLSYLKGLVTVG	bone, fracture	tartrate-resistant acid phosphatase (TRAP)	nanoparticles	[41]
4-11	SFKPSGLPAQSL	burn-injured intestine			[37,38]
CBP	LRELHLNNNC	Inflammation	Collagen	recombinant proteins	[32]

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
