# Peer review of "Systemically Administered, Target-Specific, Multi-Functional Therapeutic Recombinant Proteins in Regenerative Medicine"

_nanomaterials, 2020, doi:10.3390/nano10020226_

Round 1

Reviewer 1 Report

The review article highlights the advantages of the systemic administration of key molecules for tissue regeneration. Despite the manuscript is written concisely, several concerns are raised.

On page 3, lines 63 and 77, the authors should give examples of the side effects that can arise, at least the three more predominant.

On page 3, line 79, the authors refer antibodies, however, in the text, it is not possible to infer if they are used as therapeutic agents or targeting moieties. Moreover, the advantages of using cell-penetrating peptides instead of antibodies should be described.

CAR is extensively described in several subjects. The same should be done for other examples of peptides.

On page 14, the interest of the stem cells approach should be substantiated and more examples of stem cell studies should be given.

Author Response

On page 3, lines 63 and 77, the authors should give examples of the side effects that can arise, at least the three more predominant

Reply: We have provided the side effects that halted the human clinical trials carried on cytokines and growth factors, please see page 3, lines 61 - 65.
On page 3, line 79, the authors refer antibodies, however, in the text, it is not possible to infer if they are used as therapeutic agents or targeting moieties. Moreover, the advantages of using cell-penetrating peptides instead of antibodies should be described

Reply: We have revised the text to state clearly that we referred to therapeutic antibodies in the sentence referred above. Concerning the request to highlight the benefits of cell-penetrating peptides over antibodies, we want to state that we have described the benefits of vascular homing peptide in following sentences. Although we work on vascular homing peptides with cell-penetrating capability (i.e. believe in their potential), we do not want to present potential benefits of cell-penetrating vascular homing peptides over targeting antibodies simple because world´s top selling drugs are indeed therapeutic antibodies. Thus, the antibodies are the preferred molecule class for pharmaceutical industry and this could pave the way for targeting antibodies to clinical trials. Despite their potential, the cell-penetrating vascular homing peptides, in turn, have not made it to clinical market yet.  
CAR is extensively described in several subjects. The same should be done for other examples of peptides.

Reply: We agreed with the reviewer’s comment and devoted substantially more space for other vascular homing peptides in the field of regenerative medicine, please see highlighted sentences (new text) on pages 6, lines 186 – 199; page 8, lines 242 – 248; page 9 – 10, lines 300 – 317, . We also want to emphasize that the CAR peptide was the first vascular homing peptide described for regenerative medicine and remains as the most studied one (by publication volume) in the field to date. Thus, it naturally warrants more attention than the rest of the vascular homing peptides.  
On page 14, the interest of the stem cells approach should be substantiated and more examples of stem cell studies should be given.
Reply: We agreed with the reviewer´s assessment and we have revised the text accordingly, please see page 14, lines 458 - 463.

Reviewer 2 Report

The article entitled « Systemically administered, target specific, multi-functional therapeutic recombinant proteins in regenerative medicine” by Tero Järvinen and Toini Pemmari described how the CAR peptide technique can be applied in regenerative medicine.

This article is a review and explore potential use of local delivery of CAR peptides in vascular system for regulating angiogenesis and fibrosis.

This difficult conceptual approach is clearly described and well-illustrated (except figure 4). The limitation of this new therapeutical approach using CAR peptides in regenerating medicine is evoked. The novelty of this article is important.

Author Response

Reviewer #2:

This difficult conceptual approach is clearly described and well-illustrated (except figure 4). The limitation of this new therapeutical approach using CAR peptides in regenerating medicine is evoked. The novelty of this article is important.

Reply: We agreed with the above assessment and revised the Figure 4. In addition to that, we have provided colors for Figures # 1 and # 5

Round 2

Reviewer 1 Report

The manuscript was improved and important information was added.